# Efficient Messenger RNA Delivery to the Kidney Using Renal Pelvis Injection in Mice [note 1]

**DOI:** 10.3390/pharmaceutics13111810

**Published:** 2021-10-29

**Authors:** Natsuko Oyama, Maho Kawaguchi, Keiji Itaka, Shigeru Kawakami

**Affiliations:** 1Department of Pharmaceutical Informatics, Graduate School of Biomedical Sciences, Nagasaki University, 1-7-1 Sakamoto, Nagasaki 852-8588, Japan; n.oyama.nagasaki.u@gmail.com (N.O.); bb55319402@ms.nagasaki-u.ac.jp (M.K.); 2Department of Biofunction Research, Institute of Biomaterials and Bioengineering, Tokyo Medical and Dental University (TMDU), 2-3-10 Kanda-Surugadai, Tokyo 101-0062, Japan

**Keywords:** messenger RNA (mRNA), mRNA therapeutics, polyplex nanomicelle, kidney, renal pelvis injection, hydrodynamic injection, tubular epithelial cells

## Abstract

Renal dysfunction is often associated with the inflammatory cascade, leading to non-reversible nephrofibrosis. Gene therapy has the ability to treat the pathology. However, the difficulty in introducing genes into the kidney, via either viral vectors or plasmid DNA (pDNA), has hampered its extensive clinical use. Messenger RNA (mRNA) therapeutics has recently attracted attention as alternative gene therapies. mRNA allows protein production into post-mitotic cells without the need for transport to the nuclei in the target cells. However, few studies have reported the delivery of mRNA to the kidney. In this study, we attempted to deliver mRNA to the kidney based on the principle of pressure stimulation, by administering mRNA-loaded polyplex nanomicelles via a renal pelvis injection, directly into the kidney. Compared with the administration of naked plasmid DNA (pDNA) and naked mRNA, the mRNA-loaded nanomicelles diffusely induced protein expression in a greater number of cells at the tubular epithelium for some days. The plasma creatinine (Cre) and blood urea nitrogen (BUN) levels after the administration remained similar to those of the sham-operated controls, without marked changes in histological sections. The safety and efficacy of mRNA-loaded nanomicelles would make distinct contributions to the development of mRNA therapeutics for the kidney.

## 1. Introduction

The kidneys may be regarded as one of the most important organs for sustaining life by filtering blood, thus removing waste products amongst other reasons. Renal dysfunction first progresses to nephrofibrosis, characterized by glomerulosclerosis and tubulointerstitial fibrosis, and then irreversibly progresses to end-stage renal disease (ESRD), for which there is no curative treatment, except for renal replacement therapy (RRT) or transplantation [1]. The pathology of renal fibrosis involves inflammatory cell infiltration and fibroblast induction through the production of various growth factors, cytokines, and cell adhesion factors. Thus, it is crucial to suppress the inflammatory cascade to prevent severe exacerbation of renal function.

Gene therapy has the ability to treat multiple organs (i.e., the kidney) by providing fibrosis inhibitory factors, such as hepatocyte growth factor (HGF) [2]. However, the difficulty in introducing genes into the kidney, either via viral vectors or plasmid DNA (pDNA), has hampered its extensive clinical use [3]. mRNA therapeutics have attracted much attention as alternative gene therapies. Similar to pDNA, mRNA can encode any proteins and peptides but is internalized in the cytoplasm without the need for transport to the nuclei in the target cells, thus allowing protein production in post-mitotic cells [4]. The clinical application of mRNA vaccines during the COVID-19 pandemic revealed the safety and usefulness of mRNA as a new potential therapeutic agent [5]. However, few studies have reported the delivery of mRNA to the kidney.

In this study, we attempted to deliver mRNA to the kidney based on the principle of pressure stimulation. Several studies have attempted to introduce genes (i.e., pDNA) into the kidney using physical pressure, such as pushing [6,7] or suction [8,9]. The pressure deformed the cells, resulting in enhanced pDNA internalization into the kidney in a very short time (10 s) without causing nephrotoxicity when moderate suction pressure stimulation was used. A higher suction pressure causes nephrotoxicity in a pressure-dependent manner [9]. The pressure-stimulating gene delivery was also effective for gene transfer to the heart, where the relationship between gene expression efficiency and tissue toxicity is consistent with that of the kidney [10].

Hydrodynamic gene injection is another promising method for enhancing gene transfer by adding pressure stimulation. This technique induces fluid pressure on the target tissue by rapidly injecting a relatively large volume of injection solution, leading to extensive gene expression, regardless of tissue conditions such as fibrosis. Woodard et al. reported that hydrodynamic injection of pDNA (100 µL injected into mice over 1–3 s) from the renal pelvis allowed highly efficient gene transfer in multiple kidney cell types including glomeruli, tubules, and collecting ducts. However, these injections caused transient renal damage, as indicated by the elevation in blood urea nitrogen (BUN) level a few days after the injection, as well as the formation of a small hematoma under the kidney capsule and within the kidney parenchyma [11,12]. Recently, we also investigated hydrodynamic pDNA injection into the kidney via several local approaches from the renal infundibulum, renal artery, and renal pelvis [13]. To minimize tissue damage, we evaluated the effect of a reduced injection volume of 10 µL/mouse, together with the alteration of injection speed. Although the optimal conditions varied depending on the injection route, it was concluded that effective gene transfer was achieved by hydrodynamic injection without causing severe renal damage.

Based on our previous studies, we attempted to introduce mRNA into the kidney using the hydrodynamic method through the renal pelvis reported by Woodard et al. [11,12]. The apparent difference between pDNA and mRNA is that, although pDNA was used in the form of naked pDNA in most studies, mRNA is unlikely to be injected in the same way, owing to the very fragile nature of the mRNA. Therefore, we applied our original cationic polymer-based carrier, polyplex nanomicelles, for mRNA delivery to the kidney [14,15,16]. The nanomicelle is formed by the self-assembly of mRNA and polyethylene glycol (PEG)-polyamino acid (poly[N′-[N-(2-aminoethyl)-2-aminoethyl] aspartamide] (PAsp(DET)) block copolymers with characteristic features of precisely regulated diameters of some tens of nm, with a core-shell structure surrounded by a PEG outer shell and an mRNA-containing core for stable retention of mRNA in the carriers. Indeed, the nanomicelle exhibited excellent capacity for hydrodynamic mRNA injection to the liver [17] and muscle (under submission), as well as for smooth tissue penetration to induce protein translation diffusely around the periphery of the target site [18,19,20,21].

In this study, we administered mRNA-loaded polyplex nanomicelles via a renal pelvis injection, directly into the kidney. Naked pDNA and mRNA were used as controls. The analyses of expression profiles and safety in the kidney tissues would establish a foundation for developing new mRNA therapeutics for the treatment of kidney diseases.

## 2. Materials and Methods

### 2.1. Preparation of Plasmid DNA and Messanger RNA

pGL4.10[luc2/SV40] was purchased from Promega (Madison, WI, USA), and pZsGreen1-N1 was purchased from Clontech (Takara Bio Inc., Shiga, Japan). mRNA was prepared by in vitro transcription (IVT) using a MEGAscript T7 Transcription Kit (Ambion, Austin, TX, USA). Unmodified ribonucleic acid triphosphates were used for the IVT. The coding region of each vector was inserted into the pSP73 vector (Promega, Madison, WI, USA) for expression under the T7 promoter. To attach a poly(-A) chain to the mRNA 3 terminal, a 120-bp poly A/T sequence was cloned into the pSP73 vector downstream of the protein-coding sequence. mRNA prepared through IVT was purified using an RNeasy Mini Kit (Qiagen, Hilden, Germany). RNA was quantified by absorbance spectrophotometry using a Nanodrop 2000 spectrophotometer (Thermo Fisher Scientific, Wilmington, DE, USA). RNA quality was assessed using an Agilent 2100 Bioanalyzer chip-based capillary electrophoresis system (Agilent Technologies, Santa Clara, CA, USA).

### 2.2. Synthesis of Block Copolymers

The block copolymers were synthesized as previously reported [22]. Briefly, the polymerization of β-benzyl-L-aspartate N-carboxyanhydride (BLA-NCA) (Chuo Kasei Co. Ltd., Osaka, Japan) was initiated from the terminal primary amino group of α-methoxy-ω-amino poly (ethylene glycol) (PEG-NH_2_) (Mw 43,000) (Nippon Oil and Fats, Tokyo, Japan) to obtain PEG-b-PBLA, followed by aminolysis reaction to introduce diethylenetriamine (DET) (Wako Pure Chemical Industries, Ltd., Osaka, Japan) into the side chain of PBLA. The synthesized block polycations were determined to have a narrow unimodal molecular weight distribution (Mw/Mn = 1.04) based on gel permeation chromatography measurements. The polymerization degree of the DET segment was calculated to be 63 by ^1^H NMR analysis (JEOL EX300 spectrometer, JEOL, Tokyo, Japan).

### 2.3. Preparation of Polyplex Nanomicelles Loaded with Messenger RNA

Polyplex nanomicelles were prepared at the time of use by mixing solutions of mRNA and block copolymers (PEG-PAsp(DET)) [22]. The nanomicelle was formed through electrostatic interaction between PAsp(DET) polycations and anionic mRNA. The mRNA and block copolymers were dissolved in 10 mM HEPES buffer. The concentration of the solutions was adjusted to obtain polyplex nanomicelles with an mRNA concentration of 200 ng/μL at the N/P ratio (the residual molar ratio of the polycations amino groups to the mRNA phosphate groups) of 3. This N/P ratio was chosen because stoichiometrically charged polyplex nanomicelles were stably formed, without leaving excess polymers and mRNA molecules [23,24]. The diameter of the mRNA/PEG-PAsp(DET) nanomicelle was determined to be around 50 nm with nearly neutral surface charge [20]. The prepared mRNA polyplex solution was kept on ice until it was injected into mice.

### 2.4. Renal Pelvis Injection of Messenger RNA or Plasmid DNA

Eight-week-old male ICR mice were purchased from Japan SLC Inc. (Shizuoka, Japan). A renal pelvis injection was administered as described in the literature [11,12] with slight modifications. Mice were anesthetized with three types of mixed anesthetic agents [8] and shaved. After making an incision in the left flank, the left kidney was exposed and 10 µg of mRNA or pDNA in 50 µL of HEPES buffer was injected into the renal pelvis. The injections were administered with a 30 G 0.3 mL insulin syringe (#326638, BD Biosciences, San Jose, CA, USA) for over 8–10 s. After the needle was kept in place for 60 s, the needle was removed from the renal pelvis, and the puncture was fixed with Aron Alfa surgical adhesive (Daiichi Sankyo Co. Ltd., Tokyo, Japan).

### 2.5. In Vivo Imaging of Luciferase Activity

In vivo imaging was performed 0.25, 1, 2, 4, and 6 days after luciferase (Luc2) mRNA administration. Mice were anesthetized with isoflurane and intravenously injected with 150 mg/kg D-luciferin (#1605, Promega) in phosphate-buffered saline (PBS). After 1 min, luminescent images of the whole body were acquired using IVIS Lumina II (Caliper Life Sciences, Hopkinton, MA, USA), and total luminescence was measured in the region of interest (ROI) using Living Image 3.0 software (Caliper Life Sciences).

### 2.6. Luciferase Assay

A luciferase assay was performed as previously described [25]. Briefly, mice were sacrificed, and tissues were collected. Tissues were washed twice with cold saline and homogenized with lysis buffer. After three cycles of freeze-and-thaw cycles, homogenates were centrifuged. Luciferase activity in the transfected kidney and other tissues were normalized to the protein concentration, measured using the PicaGene (Toyobo, Osaka, Japan). The luciferase activity (ng/mg protein) of <0.001 was below the limit of quantification.

### 2.7. Immunohistochemistry

Twenty-four hours after ZsGreen1 mRNA administration, 10-µm-thick frozen sections of the kidney were prepared as described previously [26] and fixed with 4% paraformaldehyde (PFA) for 10 min. The specimens were sectioned along the coronal plane. After incubation with 1% bovine serum albumin (BSA)-PBS for 30 min at 25 °C, the sections were incubated with primary antibodies against ZsGreen (1:500 dilution, 632474; Takara Bio Inc., Shiga, Japan) and CD324 (1:100 dilution, 14-3249-82; eBioscience Inc., San Diego, CA, USA) for 16 h at 4 °C. The sections were incubated with an Alexa Fluor 488-conjugated secondary antibody (1:250 dilution, R37116; Thermo Fisher Scientific, Inc., Waltham, MA, USA) and an Alexa Fluor 647-conjugated secondary antibody (1:200 dilution, 112-605-167; Jackson Immuno Research Laboratories, Inc., West Grove, PA, USA) for 1 h at 25 °C, and reacted with 0.5 μg/mL 4-6-diamidino-2-phenylindole (DAPI; D9542; Sigma Aldrich, Inc., Saint Louis, MO, USA) in PBS for 15 min at 25 °C. The stained sections were observed under a confocal laser scanning microscope (LSM710; Carl Zeiss Microimaging GmbH, Jena, Germany).

### 2.8. Serum Creatinine and Blood Urea Nitrogen Levels

To eliminate the influence of the compensatory capacity of untreated kidneys on renal function, the right kidneys of mice were resected one week before renal pelvis injection. Blood samples were collected from the tail vein on days 1 and 7 after Luc2 mRNA administration, followed by centrifugation at 4 °C to obtain serum. Creatinine and BUN levels were measured using a DRI-CHEM NX-700 analyzer (FUJIFILM Corporation, Odawara, Japan).

### 2.9. Histomorphology Study

Twenty-four hours after Luc2 mRNA administration, the mice were perfused with PBS and 4% PFA, and the left kidneys were resected. The collected kidneys were embedded in paraffin. Paraffin-embedded sections of 5-µm thickness were stained with hematoxylin (Wako Pure Chemicals Industries, Ltd., Osaka, Japan) and eosin (Wako Pure Chemicals Industries, Ltd., Osaka, Japan) (HE). The stained sections were observed under a bright field using a fluorescence microscope, Keyence BZ-X700 (Keyence Corp., Osaka, Japan).

### 2.10. Statistical Analyses

Statistical significance was assessed using an unpaired *t*-test for two groups. Multiple comparisons were performed using Tukey’s test with analysis of variance. Statistical significance was set at *p* < 0.05.

## 3. Results

### 3.1. Efficient Messenger RNA Delivery Using Polyplex Nanomicelle via Renal Pelvis Injection

#### 3.1.1. Quantitative Measurements of Protein Expression Using Luciferase

First, mRNA or pDNA encoding Luc2 was used to quantify protein expression. Six hours after the renal pelvis injection of naked mRNA, mRNA-loaded nanomicelles, or naked pDNA, the target left kidney was excised and the protein was extracted after homogenizing the tissues. As shown in Figure 1, the mRNA groups showed greater expression than the naked pDNA. Surprisingly, even naked mRNA provided a one-order higher expression than naked pDNA, although there was no significant difference between them. The mRNA-loaded nanomicelles showed 20-fold greater expression, demonstrating rapid expression from the mRNA after injection. At the same time, other organs, such as the contralateral right kidney, liver, spleen, lung, and heart, were similarly evaluated by extracting the protein to measure Luc2 expression. We found no expression in the organs, except for a weak signal in the lungs after the injection of nanomicelles (Figure 1). Although further studies of the biodistribution of the mRNA or pDNA encoding Luc2 would be necessary to reach a conclusion, it is suggested that most of the injection bolus administered into the renal pelvis would remain at the injections site without diffusing to the circulation.

Hereafter, the time course of expression was evaluated at the target left kidney after the injection of mRNA or pDNA encoding Luc2 into the renal pelvis. For the evaluation, Luc2 expression was visualized using an in vivo imaging system (IVIS), which allowed serial imaging studies using identical mice. At 6 h after the renal pelvis injection, luminescence was detected in the target left kidney on IVIS images, as shown in the top row of Figure 2a. The photon counts indicated the expression levels with the rank order of mRNA-loaded nanomicelles > naked mRNA > naked pDNA (Figure 2b), which were in good agreement with the previous results of quantitative measurements of the Luc2 concentration in proteins (Figure 1). However, the signal of naked pDNA became slightly greater a day after administration as compared to that of the nanomicelles, although there was no significant difference between them. Thereafter, the rank order of naked pDNA > nanomicelle > naked mRNA continued until day 6 (Figure 2b). Although the signals detectable on the IVIS images mostly faded away after day 2 for each group (Figure 2a), the photon count values showed gradual decrease until day 6 (Figure 2b).

#### 3.1.2. Distribution of Expression after Injection of Messenger RNA or Plasmid DNA

The distribution of the expression in the target kidney tissue was investigated one day after the injection using mRNA or pDNA encoding the fluorescent reporter protein ZsGreen1. For mRNA-loaded nanomicelles and naked mRNA, ZsGreen1 signals were well observed. They were largely co-localized with that of anti-CD324 antibodies, indicating that the mRNA was chiefly introduced into tubular epithelial cells (Figure 3). Especially, in the medulla, the ZsGreen1 signals were observed diffusely in the tissues, which might represent the expression profile by the nanomicelles [18,19,20,21]. In contrast, after injecting naked pDNA, the number of ZsGreen1-positive cells was rather limited, but the signal intensity of each cell was brighter than that of mRNA groups. Interestingly, while the Luc2 expression, indicative of the total protein amount in the kidney, was almost comparable (without significant difference) between mRNA-loaded nanomicelles and naked pDNA on day 1 (Figure 2b), the distribution of the protein expression (Figure 3) differed markedly, typically showing different expression profiles between mRNA and pDNA.

### 3.2. Evaluation of Safety Following the Renal Pelvis Injection

#### 3.2.1. Plasma Creatinine and BUN Levels after Renal Pelvis Injection of mRNA or pDNA

Safety issues were evaluated after renal pelvis injection. As indicators of renal dysfunction, plasma creatinine (Cre) and BUN concentrations, which are commonly used as indicators of renal dysfunction, were measured at 1 and 7 days after the injection of naked DNA, naked mRNA, or mRNA-loaded polylplex nanomicelles, as well as the sham-operated mice. Although there were slight interindividual variations, there was no significant elevation of Cre and BUN levels after the injection of mRNA or pDNA compared with the sham-operated group (Figure 4). Thus, it is suggested that the injection was safely carried out, and the injection volume (50 µL) was within the tolerance limit to the renal pelvis injection.

#### 3.2.2. Histological Assessment after Renal Pelvis Injection of Messenger RNA or Plasmid DNA

The target kidney was assessed histologically 1 d after the renal pelvis injection of naked DNA, naked mRNA, or mRNA-loaded nanomicelles, as well as the kidneys of sham-operated mice (Figure 5). Compared with the sham-operated mice, there were some slight changes in the specimens of injection groups, such as tubular dilatation, hyaline casts (head arrows in Figure 5), and mononuclear infiltration (circle area in Figure 5). However, they were observed in a restricted area around the tubules, without any massive changes, such as necrosis. These results are in line with the minimal changes in blood Cre and BUN levels on day 7 (Figure 4).

## 4. Discussion

In this study, we demonstrated the administration of mRNA into the kidney by renal pelvic injection. Although the use of mRNA in the naked form could induce only a small amount of protein expression in the kidney, the incorporation of the mRNA in polyplex nanomicelles provided comparable expression with the administration of naked pDNA without inducing severe tissue damage and renal dysfunction.

An important aspect of this study is that the difference between mRNA and pDNA was represented both spatially and temporally using the same administration method in the kidney. For the temporal trends, the typical difference was the onset of the protein expression; mRNA induced the expression within some hours after the administration, whereas the expression from the pDNA was not clearly visible until day one. This reflects the intracellular mechanism leading to protein translation. pDNA needs to be transferred into the nucleus, but mRNA can be utilized for protein translation soon after being internalized into the cytoplasm. However, the time course of gradual decrease in the expression levels after day 1, revealed by serial IVIS imaging in Figure 2b, was similar for both mRNA and pDNA. This is likely due to the rapid turnover of tubular epithelial cells. Indeed, even using similar methods using polyplex nanomicelles, the duration of protein expression varied widely depending on the target organs. For example, the expression in the liver rapidly decreased within a few days [17]. In contrast, in the nervous system, the duration is somewhat longer for several days [18,27]. In skeletal muscle, the duration tends to be longer (unpublished data).

The more striking difference between mRNA and pDNA was the distribution of protein expression in the kidney tissues. As shown in Figure 3, mRNA was expressed in a diffuse manner, whereas pDNA showed different profiles with a limited number of expressing cells. This could also be attributed to the different intracellular mechanisms that lead to protein translation. When the kidney was targeted by intravascular injection [28,29,30], the mRNA may be disadvantaged due to the unstable manner. However, in this study, since the mRNA or pDNA was introduced based on the principle of pressure stimulation, their cellular uptake by passing through the plasma membrane was expected to be rather similar between them. Since the mRNA or pDNA was introduced based on the principle of pressure stimulation in this study, their cellular uptake by passing through the plasma membrane was expected to be rather similar between them. The difference in the number of expressing cells might suggest the low efficiency of the transport through the nuclear membrane to reach the nucleus. Indeed, the difficulty of introduction into the nucleus has hampered the development of non-viral DNA delivery systems [31]. In contrast, mRNA can produce proteins in the cytoplasm without the need for nucleic entry, resulting in a high ratio of expressing cells (Figure 3). However, as described in the Results section, the Luc2 measurement on day 1 revealed that the protein production in the kidney was comparable between mRNA and pDNA (Figure 2b). This result should be important when considering the therapeutic purposes; when applying mRNA or pDNA encoding secretory proteins such as growth factor, the efficacy would be theoretically comparable, or pDNA may have an advantage in the duration of protein secretion. In contrast, when the purpose is to affect as many cells as possible, mRNA has a definite therapeutic value.

In this regard, the polyplex nanomicelle made a distinctive contribution to mRNA delivery due to its high tissue penetration. This is attributed to the well-regulated particle size of several tens of nanometers, with the surface covered by dense PEG palisade [15]. Indeed, the nanomicelle could deliver mRNA to deep layers in target tissues such as joint cartilage after intra-articular injection of mRNA-loaded nanomicelles [20]. In this study, although mRNA delivery by spreading beyond the renal tubules was not clearly observed, the diffuse manner of expressing cells in the tubules was an encouraging result for revealing the potential usefulness of nanomicelles for mRNA delivery.

The safety of renal pelvic injection is one of the most critical issues for future clinical applications. Because this method uses physical forces of pressure stimulation, even though they are regulated to very low levels, it inevitably causes minor tissue damage upon injection. The important point is that the damage should be reversible without causing severe dysfunction of the target organs. BUN and Cre are the most commonly used markers of renal damage [32]. There is a correlation between these markers and histological evaluation [33,34]. The plasma Cre and BUN levels after the renal pelvis injection of any solution remained similar to those of the sham-operated group (Figure 4). In addition, tubular necrosis, which was reported by Woodard et al. [11], was not observed in the target tissues (Figure 5). These results represent the value of our refinements of injection conditions (injecting 50 µL in 8–10 s) from a previous report (100 µL in 1–3 s) [11] to reduce renal tissue damage.

In summary, we demonstrated the feasibility of using an mRNA-loaded polyplex nanomicelle for targeting the kidney based on the hydrodynamic principle. Compared with the administration of naked pDNA, the mRNA-loaded nanomicelles diffusely induced protein expression in a greater number of cells. This aspect is possibly advantageous for the treatment of renal fibrosis (partly due to tubular epithelial–mesenchymal transition) and tubular atrophy in the advanced stage of renal injury. HGF has been reported to have the potential for the repair and regeneration of renal tissues [7], but when the HGF gene was administered intramuscularly, the efficacy of HGF proteins reaching target organs from remote organs may be limited due to poor regional blood flow in the fibrotic tissues. Instead, mRNA is a promising alternative to induce HGF secretion from a wide range of tubular cells. In addition to renal fibrosis, mRNA therapeutics have widespread availability for various renal diseases with negligible risk of genotoxicity, and this study would provide useful information for the future development of mRNA therapeutics for the kidney.

## Figures and Tables

**Figure 1 pharmaceutics-13-01810-f001:**
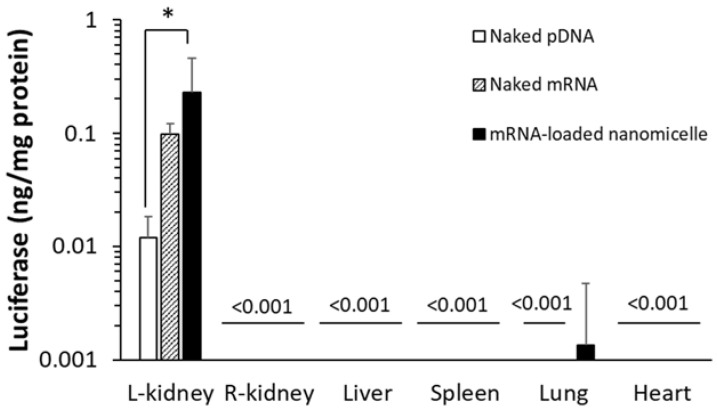
Quantitative measurements of luciferase expression by the extracted protein from each organ. Mice were injected with messenger RNA (mRNA) or plasmid DNA (pDNA) encoding Luciferase to the left kidney by renal pelvis injection. Luciferase expression levels were determined 6 h after administration. Data are represented as mean + SD (*n* = 4–6). * *p* < 0.05 (Tukey’s test).

**Figure 2 pharmaceutics-13-01810-f002:**
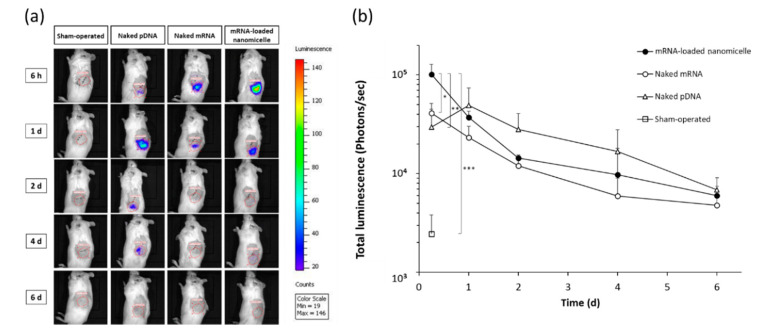
Time course of luciferase expression in the left kidney after renal pelvis injection of messenger RNA (mRNA) or plasmid DNA (pDNA) before in vivo imaging was performed. (**a**) Representative IVIS images of mice at 6 h, 1, 2, 4, and 6 d after renal pelvis injection of naked pDNA, naked mRNA, or mRNA-loaded polyplex nanomicelles. (**b**) Quantification of luciferase expression was performed with the total luminescence (photons/sec) in the left kidney was determined by same size of ROI. Data are represented as mean + SD (*n* = 3–4). * *p* < 0.05, ** *p* < 0.01, *** *p* < 0.001 (Tukey’s test).

**Figure 3 pharmaceutics-13-01810-f003:**
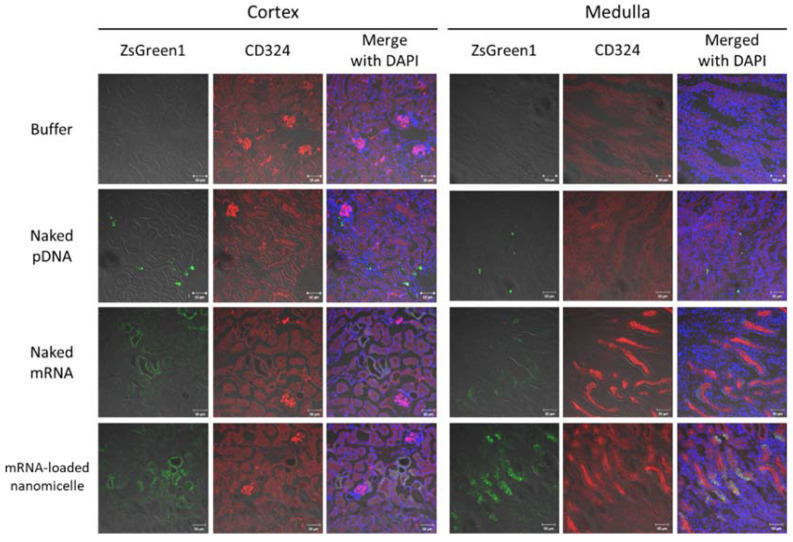
Distribution of ZsGreen1 expression in the kidney following renal pelvis injection. Mice were injected with ZsGreen1 messenger RNA or plasmid DNA by renal pelvis injection. At 24 h after injection, the kidney tissues were histologically analyzed with anti-ZsGreen1 antibody and CD324 (specified for tubular epithelial cells)-antibody staining. The stained sections were observed by confocal laser scanning microscopy. Objective lens: ×40 lens. Green: ZsGreen1 expression; Red: CD324; Blue: DAPI. Scale bars represent 50 µm.

**Figure 4 pharmaceutics-13-01810-f004:**
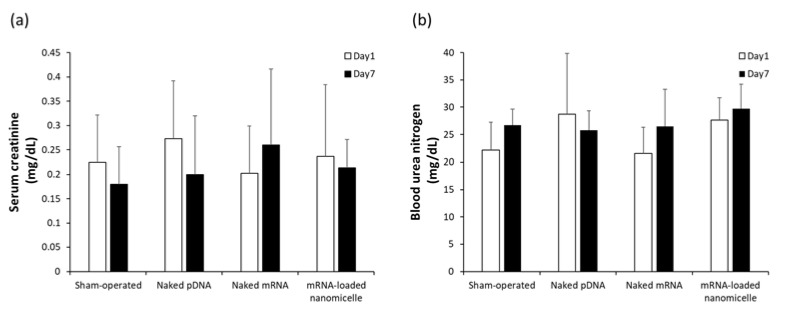
(**a**) Serum creatinine (Cre) and (**b**) Blood Urea Nitrogen (BUN) levels after renal pelvis injection of messenger RNA (mRNA) or plasmid DNA (pDNA). The blood was collected on day 1 and day 7 after the injection of naked pDNA, naked mRNA (Luc2), or mRNA-loaded polylplex nanomicelles. Serum Cre and BUN levels were measured using a DRI-CHEM NX-700 analyser. Data are represented as mean + SD (*n* = 4–5).

**Figure 5 pharmaceutics-13-01810-f005:**
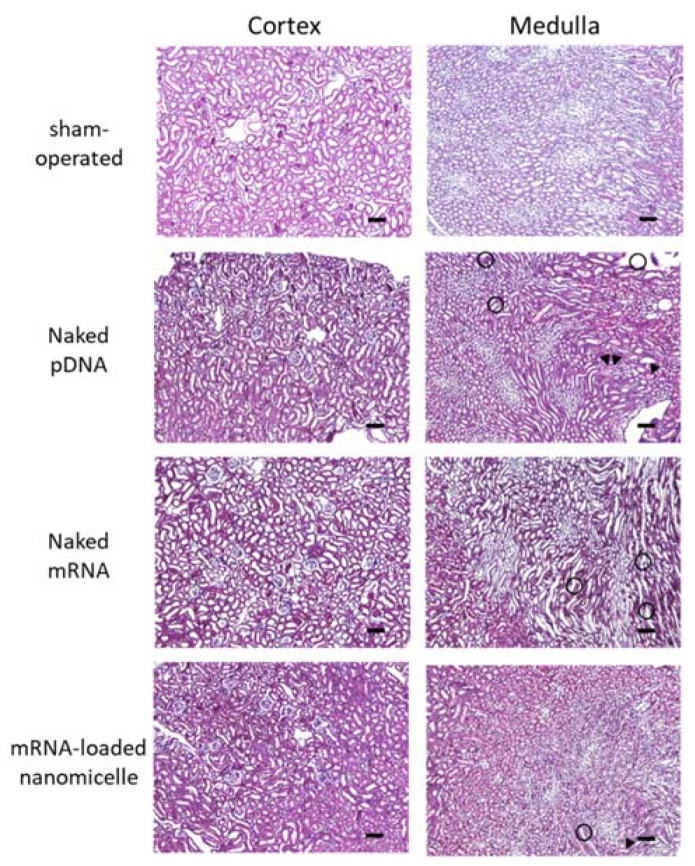
Histological assessment after renal pelvis injection of messenger RNA (mRNA) or plasmid DNA (pDNA). Mice were injected with luciferase (Luc2) mRNA or pDNA by renal pelvis injection. The kidneys were resected 24 h after the injection, followed by histological analysis of 4 µm paraffin sections prepared with hematoxylin and eosin staining. The kidneys of sham-operated mouse were also assessed as a negative control. Objective lens: ×10 lens. Scale bars represent 100 µm. head arrow, hyaline casts; circle area, mononuclear infiltration.

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
