# Peer review of "Efficient Messenger RNA Delivery to the Kidney Using Renal Pelvis Injection in Mice"

_pharmaceutics, 2021, doi:10.3390/pharmaceutics13111810_

Round 1

Reviewer 1 Report

Comments to authors:

This article presents an exploration of a mRNA administration technique combined with pressure stimulation to induce protein production in the kidneys. The authors compare the efficacy of pressure stimulation delivery of naked mRNA and naked pDNA with mRNA that is encapsulated in a nanomicelle carrier. The use of mRNA is a promising advancement in gene therapy as it does not require translocation of the construct to the nucleus of the cells and does not permanently alter the DNA of the target tissue.The authors find that their mRNA is able to produce a robust expression of luciferase protein that remains localized to the kidney region, and is detectable for a period of several days. To evaluate the distribution of protein induction in the kidney, the authors next use a green fluorescent protein construct and use histology to suggest that there is a higher level of expression of the micelle mRNA throughout the cortex and medulla of the kidneys. In prior work, the authors found damage to the kidneys with pressure stimulation, which appears to be abrogated in the current study by reducing the injection volume (as measured by serum creatinine, blood urea nitrogen, and histological evaluation). The findings are relatively well supported, and related studies examining administration strategies for DNA/RNA have been published in Pharmaceutics in the past. Therefore, in the eyes of this reviewer the content is most likely to be of interest for the readership of Pharmaceutics, and the data presented is similarly thorough. Prior to acceptance there are a few comments that need to be addressed as detailed below, therefore this reviewer suggests major revision.

Major comments:

-The authors mention that they use the “principle of pressure stimulation” and “introduce mRNA into the kidney via the renal pelvis” It is not clear to this reviewer what exactly this means, and how similar or different the approach is from the other approaches the authors discuss in the introduction. Please include a figure or sub-panel of a figure that has a diagram of the procedure  for clarification.  

-Line 185—> “First, mRNA or pDNA encoding Luc2 was used to quantify protein expression” How is this procedure done? Although there is a reference provided in the methods please provide at least some description, also need to indicate the LOD/LOQ in the text and figure. Please denote that rather than “<0.001” in the figure.

-Lines 215-218—>The authors mention “Although the signals detectable on the IVIS images mostly faded away after day 2 for each group (Fig. 2a), the photon count values were still higher than those of sham-operated mice even on day 6 (Fig. 2b), suggesting that a very small quantity of expression might be sustained until the date.” If the difference was statistically significant, please note the p value and test used. If it was not, the photon values cannot be said to be higher and so it is suggested to alter the phrasing to be in keeping with the statistics.

-Lines 230-232—>”For mRNA-loaded nanomicelles and naked mRNA, ZsGreen1 signals were diffusely observed in the tissues and largely colocalized with that of anti-CD324 antibodies, indicating that the mRNA was chiefly introduced into tubular epithelial cells (Fig. 3)” In the medulla this reviewer would agree with the suggestion of colocalization, but the cortex images appear to be somewhat different. Please expand this description to match the observations in the figure.

-Lines 271-273—> the authors mention “Thus, although a slight degree of tissue damage might have been induced by the injection, it was probably transient and reversible by considering the results of minimal changes in blood Cre and BUN levels on day 7” Please rephrase to keep the description of the results out of the realm of speculation- this evidence does indeed suggest minor damage but doesn’t say anything about transience or reversibility.

-Lines 296-297—> the authors mention here and elsewhere the “target tubular epithelial cells.” Why are these the target? In what way are they targeted versus other cells in the area? It is suggested to either support the statement or to simply describe the findings and possible implications both here and throughout the manuscript. If the tubular epithelial cells happen to show higher efficiency of expression, and would therefore make good targets for subsequent protein delivery, then just say that instead.

-Lines 306-307—> Although it was mentioned in the introduction, further discussion of the pressure stimulation vs other administration techniques would strengthen this section.

Minor comments:

-Line 118, “2.3 Preparation of polyplex nanomicelles loaded with messagner RNA” typo

-Line 183, “3.1. Efficient messanger RNA” —> “3.1. Efficient messenger RNA”  

-Markers of kidney damage are outside of this reviewer’s expertise, so additional text in the discussion describing the relative acceptance of the creatinine, blood urea nitrogen, and histology versus other techniques would be helpful for a more broad readership. 

Author Response

For Expert Reviewer 1

We are grateful to Reviewer 1 for the critical comments and useful suggestions that have helped us to improve our paper. As indicated in the responses that follow, we have taken all these comments and suggestions into account in the revised version of our paper.

Comment 1:

-The authors mention that they use the “principle of pressure stimulation” and “introduce mRNA into the kidney via the renal pelvis” It is not clear to this reviewer what exactly this means, and how similar or different the approach is from the other approaches the authors discuss in the introduction. Please include a figure or sub-panel of a figure that has a diagram of the procedure for clarification.

Response:

Thank you for your suggestion. In this study, we attempted to introduce mRNA into the kidney using the same procedure as the hydrodynamic method through the renal pelvis reported by Woodard et al [11, 12]. However, the explanation was not sufficient. We have cited these papers [11, 12] again with a specific description (Lines 76-77).

(Original)

Based on previous studies, we attempted to introduce mRNA into the kidney via the renal pelvis. (Lines 76-77)

(Revise)

Based on our previous studies, we attempted to introduce mRNA into the kidney using the hydrodynamic method through the renal pelvis reported by Woodard et al [11, 12]. (Lines 76-78)

Comment 2:

-Line 185—> “First, mRNA or pDNA encoding Luc2 was used to quantify protein expression” How is this procedure done? Although there is a reference provided in the methods please provide at least some description, also need to indicate the LOD/LOQ in the text and figure. Please denote that rather than “<0.001” in the figure.

Response:

We have added an explanation on how to measure luciferase activity in vivo. In addition, we have added the explanation that the luciferase activity (ng/mg protein) of <0.001 was below the limit of quantification was also added.

(Original)

A luciferase assay was performed as previously described [25]. Luciferase activity in the transfected kidney was normalized to the protein concentration, measured using the PicaGene (Toyobo, Osaka, Japan). (Lines 145-147)

(Revise)

A luciferase assay was performed as previously described [25]. Briefly, mice were sacrificed and tissues were collected. Tissues were washed twice with cold saline and homogenized with lysis buffer. After three cycles of freeze-and-thaw cycles, homogenates were centrifuged. Luciferase activity in the transfected kidney and other tissues were normalized to the protein concentration, measured using the PicaGene (Toyobo, Osaka, Japan). The luciferase activity (ng/mg protein) of <0.001 was below the limit of quantification. (Lines 149-155)

Comment 3:

-Lines 215-218—>The authors mention “Although the signals detectable on the IVIS images mostly faded away after day 2 for each group (Fig. 2a), the photon count values were still higher than those of sham-operated mice even on day 6 (Fig. 2b), suggesting that a very small quantity of expression might be sustained until the date.” If the difference was statistically significant, please note the p value and test used. If it was not, the photon values cannot be said to be higher and so it is suggested to alter the phrasing to be in keeping with the statistics.

Response:

Thank you for pointing this out. We agree the statement the reviewer pointed out was inadequate without statistical basis. We revised the sentence in Lines 215-218 in the original manuscript to correctly state the results.

(Original)

Although the signals detectable on the IVIS images mostly faded away after day 2 for each group (Fig. 2a), the photon count values were still higher than those of sham-operated mice even on day 6 (Fig. 2b), suggesting that a very small quantity of expression might be sustained until the date.

(Revise)

Although the signals detectable on the IVIS images mostly faded away after day 2 for each group (Fig. 2a), the photon count values showed gradual decrease until day 6 (Fig. 2b). (Lines 223-225)

Comment 4:

-Lines 230-232—>”For mRNA-loaded nanomicelles and naked mRNA, ZsGreen1 signals were diffusely observed in the tissues and largely colocalized with that of anti-CD324 antibodies, indicating that the mRNA was chiefly introduced into tubular epithelial cells (Fig. 3)” In the medulla this reviewer would agree with the suggestion of colocalization, but the cortex images appear to be somewhat different. Please expand this description to match the observations in the figure.

Response:

Thank you for suggesting an important point. We intensely observed the images again. We believe that the diffuse manner of expression is an important aspect of mRNA-loaded nanomicelles, and we really found this trend in the images of medulla. However, we agree it is difficult to conclude it by the images of cortex. To state the observation precisely, we revised the description as follows.

(Original)

For mRNA-loaded nanomicelles and naked mRNA, ZsGreen1 signals were diffusely observed in the tissues and largely colocalized with that of anti-CD324 antibodies, indicating that the mRNA was chiefly introduced into tubular epithelial cells (Fig. 3). (Lines 230-232)

(Revise)

For mRNA-loaded nanomicelles and naked mRNA, ZsGreen1 signals were well observed. They were largely colocalized with that of anti-CD324 antibodies, indicating that the mRNA was chiefly introduced into tubular epithelial cells (Fig. 3). Especially, in medulla, the ZsGreen1 signals were observed diffusely in the tissues, which might represent the expression profile by the nanomicelles [18-21]. (Lines 237-241)

Comment 5:

-Lines 271-273—> the authors mention “Thus, although a slight degree of tissue damage might have been induced by the injection, it was probably transient and reversible by considering the results of minimal changes in blood Cre and BUN levels on day 7” Please rephrase to keep the description of the results out of the realm of speculation- this evidence does indeed suggest minor damage but doesn’t say anything about transience or reversibility.

Response:

Thank you for your suggestion. We revised the sentences to correctly address the results as follows:

(Original)

Although there were slight interindividual variations, there was no significant elevation of Cre and BUN levels after the injection of mRNA or pDNA compared with the sham-operated group. (Lines 252-254)

(Revise)

Although there were slight interindividual variations, there was no significant elevation of Cre and BUN levels after the injection of mRNA or pDNA compared with the sham-operated group (Fig. 4). (Lines 261-263)

(Original)

However, they were observed in a restricted area around the tubules, without any massive changes, such as necrosis. Thus, although a slight degree of tissue damage might have been induced by the injection, it was probably transient and reversible by considering the results of minimal changes in blood Cre and BUN levels on day 7 (Fig. 4). (Lines 271-273)

(Revise)

However, they were observed in a restricted area around the tubules, without any massive changes, such as necrosis. These results are in line with the minimal changes in in blood Cre and BUN levels on day 7 (Fig. 4). (Lines 278-281)

Comment 6:

-Lines 296-297—> the authors mention here and elsewhere the “target tubular epithelial cells.” Why are these the target? In what way are they targeted versus other cells in the area? It is suggested to either support the statement or to simply describe the findings and possible implications both here and throughout the manuscript. If the tubular epithelial cells happen to show higher efficiency of expression, and would therefore make good targets for subsequent protein delivery, then just say that instead.

Response:

Thank you for your suggestion. As you pointed out, we did not actively target the epithelial cells, but accordingly, the expression was chiefly observed in the epithelial cells. To avoid misinterpretation, we would like to omit the word of "target.

(Original)

This is likely due to the rapid turnover of target tubular epithelial cells. (Lines 296)

(Revise)

This is likely due to the rapid turnover of tubular epithelial cells. (Lines 304)

Comment 7:

-Lines 306-307—> Although it was mentioned in the introduction, further discussion of the pressure stimulation vs other administration techniques would strengthen this section.

Response:

Following your suggestion, we have added explanations on pDNA delivery and mRNA delivery by pressure stimulation.

(Revise)

The more striking difference between mRNA and pDNA was the distribution of protein expression in the kidney tissues. As shown in Fig. 3, mRNA was expressed in a diffuse manner, whereas pDNA showed different profiles with a limited number of expressing cells. This could also be attributed to the different intracellular mechanisms that lead to protein translation. When the kidney was targeted by intravascular injection [28, 29, 30], the mRNA may be disadvantaged due to the unstable manner. However, in this study, since the mRNA or pDNA was introduced based on the principle of pressure stimulation, their cellular uptake by passing through the plasma membrane was expected to be rather similar between them. Since the mRNA or pDNA was introduced based on the principle of pressure stimulation in this study, their cellular uptake by passing through the plasma membrane was expected to be rather similar between them. (Lines 310-320)

Reference

[28] Tomita, N.; Higaki, J.; Morishita, R.; Kato, K.; Mikami, H.; Kaneda, Y.; Ogihara, T. Directed in vivo gene introduction into rat kidney. Biochem. Biophys. Res. Commun. 1992, 186 (1), 129-134, doi.org/10.1016/S0006-291X(05)80784-3

[29] Lai L. W.; Moeckel G. W.; Lien Y. H. Kidney-targeted liposome-mediated gene transfer in mice. Gene Ther. 1997, 4, 426-431, doi.org/10.1038/sj.gt.3300406 doi: 10.1038/sj.gt.3300406

[30] Moullier, P.; Friedlander, G.; Calise, D.; Ronco, P.; Perricaudet, M.; Ferry, N. Adenoviral-mediated gene transfer to renal tubular cells in vivo. Kidney Int. 1994, 45 (4), 1220-1225, doi.org/10.1038/ki.1994.162

Comment 8:

-Line 118, “2.3 Preparation of polyplex nanomicelles loaded with messagner RNA” typo

Response:

Thank you for pointing out. We corrected as follows.

(Original)

2.3 Preparation of polyplex nanomicelles loaded with messagner RNA

(Revise)

2.3 Preparation of polyplex nanomicelles loaded with messenger RNA

Comment 9:

-Line 183, “3.1. Efficient messanger RNA” —> “3.1. Efficient messenger RNA” 

Response:

Thank you for pointing out. We corrected as follows.

(Original)

3.1. Efficient messanger RNA delivery using polyplex nanomicelle via renal pelvis injection

(Revise)

3.1. Efficient messenger RNA delivery using polyplex nanomicelle via renal pelvis injection

Comment 10:

-Markers of kidney damage are outside of this reviewer’s expertise, so additional text in the discussion describing the relative acceptance of the creatinine, blood urea nitrogen, and histology versus other techniques would be helpful for a more broad readership.

Response:

Thank you for your suggestion of evaluation of the other kidney damage markers.

We add the following text in the discussion to describe the markers.

(Revise)

As indicators of renal dysfunction, plasma creatinine (Cre) and BUN concentrations, which are commonly used as indicators of renal dysfunction, were measured at 1 and 7 days after the injection of naked DNA, naked mRNA, or mRNA-loaded polylplex nanomicelles, as well as the sham-operated mice. (Lines 257-261)

The safety of renal pelvic injection is one of the most critical issues for future clinical applications. Because this method uses physical forces of pressure stimulation, even though they are regulated to very low levels, it inevitably causes minor tissue damage upon injection. The important point is that the damage should be reversible without causing severe dysfunction of the target organs. To evaluate the damage, we measured the plasma creatinine and BUN levels, which are the most commonly-used markers of renal damage [34]. These markers are also known to be associated with the histopathologic features [35. 36]. As shown in Fig. 4, the plasma Cre and BUN levels after the renal pelvis injection of any solution remained similar to those of the sham-operated group (Fig. 4). (Lines 340-347)

Reference

[33] Fuchs, T. C. ; Hewitt, P. Biomarkers for Drug-Induced Renal Damage and Nephrotoxicity—An Overview for Applied Toxicology. The AAPS Journal, 2011, 13 (4), 615-631. doi: 10.1208/s12248-011-9301-x

[34] Qi, Z.; Li, Z.; Li, W.; Liu, Y.; Wang, C.; Lin, H.; Liu, J.; Li, P. Pseudoginsengenin DQ Exhibits Therapeutic Effects in Cisplatin-Induced Acute Kidney Injury via Sirt1/NF-κB and Caspase Signaling Pathway without Compromising Its Antitumor Activity in Mice. Molecules 2018, 23(11), 3038. doi.org/10.3390/molecules23113038.

[35] Wei, Q.; Dong, G.; Chen, J.; Ramesh, G.; Dong, Z. Bax and Bak have critical roles in ischemic acute kidney injury in global and proximal tubule–specific knockout mouse models. Kidney Int. 2013, 84, 138–148. doi:10.1038/ki.2013.6

Reviewer 2 Report

The manuscript by Oyama et al., evaluates the effects of mRNA-loaded polyplex nanomicelles delivery to rodent kidneys. Indeed, this is an interesting study which provided some interesting findings. They have assessed the mRNA delivery efficiency, Plasma creatinine and BUN levels after renal pelvis injection and morphological analysis of after pelvic kidney after renal pelvis injection of mRNA.

However, i think the safety profile of the study can be further validated through some other experiments such as assessing the levels of stress kinases (JNK…), superoxide dismutase, catalase, malondialdehyde etc.

Author Response

For Expert Reviewer 2

We are grateful to Reviewer 2 for the critical comments and useful suggestions that have helped us to improve our paper. As indicated in the responses that follow, we have taken all these comments and suggestions into account in the revised version of our paper.

Comment:

The manuscript by Oyama et al., evaluates the effects of mRNA-loaded polyplex nanomicelles delivery to rodent kidneys. Indeed, this is an interesting study which provided some interesting findings. They have assessed the mRNA delivery efficiency, Plasma creatinine and BUN levels after renal pelvis injection and morphological analysis of after pelvic kidney after renal pelvis injection of mRNA.

However, i think the safety profile of the study can be further validated through some other experiments such as assessing the levels of stress kinases (JNK…), superoxide dismutase, catalase, malondialdehyde etc.

Response:

Thank you very much for the suggestive comments. We have started the first research on whether mRNA or mRNA-loaded nanomicelles and pressure stimulation in the kidney will increase protein expression, where the expressing cells are located, and whether toxicity will occur. In this study, we evaluated plasma Cre and BUN levels, which are commonly used markers of renal damages, and assessed histopathological features by using hematoxylin eosin staining. These markers are also known to be associated with the histopathologic features. To address the methods for evaluating the nephrotoxicity rationally, we added some sentences in the main text as follows.

We agree other markers such as stress kinases, you kindly suggested, would provide further insights into the nephrotoxicity. We would like to include the markers for the evaluation, especially when investigating the therapeutic efficacy of the mRNAs using animal disease models.

(Revise)

As indicators of renal dysfunction, plasma creatinine (Cre) and BUN concentrations, which are commonly used as indicators of renal dysfunction, were measured at 1 and 7 days after the injection of naked DNA, naked mRNA, or mRNA-loaded polylplex na-nomicelles, as well as the sham-operated mice. (Lines 257-261)

The safety of renal pelvic injection is one of the most critical issues for future clinical applications. Because this method uses physical forces of pressure stimulation, even though they are regulated to very low levels, it inevitably causes minor tissue damage upon injection. The important point is that the damage should be reversible without causing severe dysfunction of the target organs. To evaluate the damage, we measured the plasma creatinine and BUN levels, which are the most commonly-used markers of renal damage [34]. These markers are also known to be associated with the histopathologic features [35. 36]. As shown in Fig. 4, the plasma Cre and BUN levels after the renal pelvis injection of any solution remained similar to those of the sham-operated group (Fig. 4). (Lines 340-347)

Reference

[33] Fuchs, T. C. ; Hewitt, P. Biomarkers for Drug-Induced Renal Damage and Nephrotoxicity—An Overview for Applied Toxicology. The AAPS Journal, 2011, 13 (4), 615-631. doi: 10.1208/s12248-011-9301-x

[34] Qi, Z.; Li, Z.; Li, W.; Liu, Y.; Wang, C.; Lin, H.; Liu, J.; Li, P. Pseudoginsengenin DQ Exhibits Therapeutic Effects in Cisplatin-Induced Acute Kidney Injury via Sirt1/NF-κB and Caspase Signaling Pathway without Compromising Its Antitumor Activity in Mice. Molecules 2018, 23(11), 3038. doi.org/10.3390/molecules23113038.

[35] Wei, Q.; Dong, G.; Chen, J.; Ramesh, G.; Dong, Z. Bax and Bak have critical roles in ischemic acute kidney injury in global and proximal tubule–specific knockout mouse models. Kidney Int. 2013, 84, 138–148. doi:10.1038/ki.2013.6

Reviewer 3 Report

The authors presented a manuscript comparing naked pDNA, naked mRNA, and mRNA-loaded polyplex nanomicelles for protein expression and distribution via a renal pelvis injection into mice. The results demonstrated safety and improved expression for mRNA-loaded polyplex nanomicelles. However, necessary characterizations of mRNA-loaded polyplex nanomicelles needs to be provided to convince broad readers.

Additional comments:

1) how was unencapsulated mRNA removed from nanomicelles?

2) mRNA encapsulation efficiency or concentration in polyplex nanomicelles?

3) Size of the mRNA-loaded polyplex nanomicelles?

Author Response

For Expert Reviewer 3

We are grateful to Reviewer 3 for the critical comments and useful suggestions that have helped us to improve our paper. As indicated in the responses that follow, we have taken all these comments and suggestions into account in the revised version of our paper.

Comment 1:

how was unencapsulated mRNA removed from nanomicelles?

Response:

As described in 2.3 in the original manuscript, the nanomicelle is formed by mixing mRNA solutions and block copolymers. The driving force of micelle formation is the electrostatic interaction between anionic mRNA and cationic polymers. Then, after internalized in the cells, the mRNA is considered to be released through an interexchange reaction with counter polyanions in the cytoplasm.

We previously investigated the release kinetics in detail using fluorescent-labeled pDNA-loaded nanomicelles, in which the pDNA was gradually released after internalized in the cytoplasm (Itaka K, et al. J Gene Med 2004; 6: 76-84). Unfortunately, we didn’t use mRNA for the analyses. However, since the electrical property should be similar mRNA and pDNA, we believe that mRNA would be gradually released through the similar mechanism in the cells.

Comment 2:

mRNA encapsulation efficiency or concentration in polyplex nanomicelles?

Response:

When forming nanomicelles, we usually use excess amount of polycations to anionic mRNA. As mentioned in the previous response, the nanomicelle is formed by the electrostatic interaction between anionic mRNA and cationic polymers. Theoretically, in the presence of excess amount of polycations, all mRNA molecules would be trapped by the polymers.

In this study, we used the condition of N/P ratio (the residual molar ratio of the polycations amino groups to the mRNA phosphate groups) of 3, as described in 2.3. Because half of the amino groups in PAsp(DET) are protonated at pH=7.4, the ratio for forming stoichiometric polyplexes is theoretically N/P = 2 (N+/P = 1) (referred as [22] in the original manuscript). However, by evaluating the nanomicelles with various N/P ratios, we found that the physicochemical properties levelled off over N/P ≈ 3, not N/P = 2 (referred as [23, 24] in the original manuscript). Thus, we chose the condition of N/P = 3 in this study, by which we can expect to prepare the stoichiometric charged polyplexes without leaving excess polymers and mRNA molecules.

Comment 3:

Size of the mRNA-loaded polyplex nanomicelles?

Response:

The size of mRNA-loaded polyplex nanomicelles was analyzed in our previous study (referred as [20] in the original manuscript). By dynamic light scattering measurement, the size of the mRNA/PEG-PAsp(DET) nanomicelle was 52.83 ± 1.4 nm (polydispersity (PDI) = 0.162 ± 0.019), with the zeta potential of 0.071 ± 0.159 mV.

To address the features suggested by the reviewer (comments 1-3), we would like to revise the section of 2.3 Preparation of polyplex nanomicelles loaded with messegner RNA, as follows.

(Original)

2.3 Preparation of polyplex nanomicelles loaded with messagner RNA

Polyplex nanomicelles were prepared at the time of use by mixing solutions of mRNA and block copolymers (PEG-PAsp(DET)). The mRNA and block copolymers were dissolved in 10 mM HEPES buffer. The concentration of the solutions was adjusted to ob-tain polyplex nanomicelles with an mRNA concentration of 200 ng/μL at the N/P ratio (the residual molar ratio of the polycations amino groups to the mRNA phosphate groups) of 3. This N/P ratio was chosen because stoichiometrically charged polyplex na-nomicelles were stably formed at N/P ≥ 3 [23, 24]. The prepared mRNA polyplex solution was kept on ice until it was injected into mice.

(Revise)

Polyplex nanomicelles were prepared at the time of use by mixing solutions of mRNA and block copolymers (PEG-PAsp(DET)) [22]. The nanomicelle was formed through electrostatic interaction between PAsp(DET) polycations and anionic mRNA. The mRNA and block copolymers were dissolved in 10 mM HEPES buffer. The concentration of the solutions was adjusted to obtain polyplex nanomicelles with an mRNA concentration of 200 ng/μL at the N/P ratio (the residual molar ratio of the polycations amino groups to the mRNA phosphate groups) of 3. This N/P ratio was chosen because stoichiometrically charged polyplex nanomicelles were stably formed, without leaving excess polymers and mRNA molecules [23, 24]. The diameter of the mRNA/PEG-PAsp(DET) nanomicelle was determined to be around 50 nm with nearly neutral surface charge [20]. The prepared mRNA polyplex solution was kept on ice until it was injected into mice. (Lines 120-130)

Round 2

Reviewer 3 Report

The authors have addressed my concerns.

Author Response

For Expert Reviewer 3

We are grateful to Reviewer 3 for your kind comments. As indicated in the responses that follow, we carefully reviewed the manuscript and corrected the following spelling errors into the revised version of our manuscript.

Response:

Thank you for your kind comments about spell check. We corrected the following six spelling errors.

(Original) (Lines 108-115)

2.2. Synthesis of block copolymers

Briefly, the polymerization of β-benzyl-L-aspartate N-carboxyanhydride (BLA-NCA) (Chuo Kasei Co. Ltd. Osaka, Japan) was initiated from the terminal primary amino group of α-methoxy-ω-amino poly (ethylene glycol) (PEG-NH2) (Mw 43,000) (Nippon Oil and Fats, Tokyo, Japan) to obtain PEG-b-PBLA, followed by aminolysis reaction to introduce diethylenetriamine (DET) (Wako Pure Chemical Industries, Ltd., Osaka, Ja-pan) into the side chain of PBLA.

(Revise)

2.2. Synthesis of block copolymers

Briefly, the polymerization of β-benzyl-L-aspartate N-carboxyanhydride (BLA-NCA) (Chuo Kasei Co. Ltd. Osaka, Japan) was initiated from the terminal primary amino group of α-methoxy-ω-amino poly (ethylene glycol) (PEG-NH2) (Mw 43,000) (Nippon Oil and Fats, Tokyo, Japan) to obtain PEG-b-PBLA, followed by aminolysis reaction to introduce diethylenetriamine (DET) (Wako Pure Chemical Industries, Ltd., Osaka, Ja-pan) into the side chain of PBLA. (Lines 109-115)

(Original) (Lines 115-118)

The synthesized block polycations were determined to have a narrow unimodal mo-lecular weight distribution (Mw/Mn = 1.04) based on gel permeation chromatography measurements. The polymerization degree of the DET segment was calculated to be 63 by 1H NMR analysis (JEOL EX300 spectrometer, JEOL, Tokyo, Japan).

(Revise)

The synthesized block polycations were determined to have a narrow unimodal mo-lecular weight distribution (Mw/Mn = 1.04) based on gel permeation chromatography measurements. The polymerization degree of the DET segment was calculated to be 63 by 1H NMR analysis (JEOL EX300 spectrometer, JEOL, Tokyo, Japan). (Lines 115-118)

(Original) (Lines 131)

2.4. Renal pelvis injection of messanger RNA or plasmid DNA

(Revise)

2.4. Renal pelvis injection of messenger RNA or plasmid DNA (Lines 131)

(Original) (Lines 171-173)

2.8. Serum Creatinine and Blood Urea Nitrogen levels

To eliminate the influence of the compensatory capacity of untreated kidneys on renal function, the right kidneys of mice were resected ne week before renal pelvis injection.

(Revise)

2.8. Serum Creatinine and Blood Urea Nitrogen levels

To eliminate the influence of the compensatory capacity of untreated kidneys on renal function, the right kidneys of mice were resected one week before renal pelvis injection.

(Original) (Lines 186-189)

2.10. Statistical analyses

Statistical significance was assessed using an unpaired t-test for two groups. Mul-tiple comparisons were performed using Tukey’s test with analysis of variance. Statis-tical significance was set at p<0.05.

(Revise)

2.10. Statistical analyses

Statistical significance was assessed using an unpaired t-test for two groups. Mul-tiple comparisons were performed using Tukey’s test with analysis of variance. Statis-tical significance was set at P<0.05.

(Original) (Lines 208-211)

Figure 1. Quantitative measurements of luciferase expression by the extracted protein from each organ. Mice were injected with messenger RNA (mRNA) or plasmid DNA (pDNA) encoding Luciferase to the left kidney by renal pelvis injection. Luciferase expression levels were deter-mined 6 h after administration. Data are represented as mean + SD (n=4-6). *P<0.05 (Tukey’s test).

(Revise)

Figure 1. Quantitative measurements of luciferase expression by the extracted protein from each organ. Mice were injected with messenger RNA (mRNA) or plasmid DNA (pDNA) encoding Luciferase to the left kidney by renal pelvis injection. Luciferase expression levels were deter-mined 6 h after administration. Data are represented as mean + SD (n=4-6). *P<0.05 (Tukey’s test).
